# Genome-wide analyses of neonatal jaundice reveal a marked departure from adult bilirubin metabolism

Pol Solé-Navais [1,27] ✉, Julius Juodakis[1,27], Karin Ytterberg[1], Xiaoping Wu [2,3], Jonathan P. Bradfield[4,5], Marc Vaudel [6,7], Abigail L. LaBella [8], Øyvind Helgeland [6,7], Christopher Flatley [1], Frank Geller [2,3], Moshe Finel[9], Mengqi Zhao [10], Philip Lazarus[10,26], Hakon Hakonarson [4,11,12,13], Per Magnus [14], Ole A. Andreassen [15,16,17], Pål R. Njølstad [6,18], Struan F. A. Grant [4,11,12,19,20], Bjarke Feenstra [2,3,21], Louis J. Muglia [22,23,24], Stefan Johansson [6,25], Ge Zhang[23,24] & Bo Jacobsson [1,7] ✉

Jaundice affects almost all neonates in their first days of life and is caused by the accumulation of bilirubin. Although the core biochemistry of bilirubin metabolism is well understood, it is not clear why some neonates experience more severe jaundice and require treatment with phototherapy. Here, we present the first genome-wide association study of neonatal jaundice to date in nearly 30,000 parent-offspring trios from Norway (cases ≈ 2000). The alternate allele of a common missense variant affecting the sequence of *UGT1A4* reduces the susceptibility to jaundice five-fold, which replicated in separate cohorts of neonates of African American and European ancestries. eQTL colocalization analyses indicate that the association may be driven by regulation of *UGT1A1* in the intestines, but not in the liver. Our results reveal marked differences in the genetic variants involved in neonatal jaundice compared to those regulating bilirubin levels in adults, suggesting distinct genetic mechanisms for the same biological pathways.

Most neonates are affected by jaundice in their first days of life, caused by accumulation of bilirubin (hyperbilirubinemia). While this condition occurs to some degree in 50–90% of births, usually it resolves without treatment[1,2]. Phototherapy is used in more serious cases, around 5–10% of newborns, and can be effective[3]. Yet, severe cases of jaundice still occur—leading to neurotoxicity, long-term cognitive impairment, and in rare cases kernicterus and death—and cause a significant burden on global health[4–6].

Many questions about the etiology of neonatal jaundice remain, although the core biochemistry of bilirubin metabolism is known. Bilirubin (*unconjugated* form) is produced in the breakdown of hemoglobin, then taken up into hepatocytes, and converted to *conjugated* bilirubin by uridine diphosphate-glucuronosyltransferase (UGT)[7,8]. Conjugated bilirubin is transported to bile, while hopping in

and out of hepatocytes to prevent local accumulation[9], then travels to intestines for excretion via feces[7]. In neonates, liver UGT expression is low, while the production of unconjugated bilirubin is high (because of faster turnover of red blood cells and fetal hemoglobin), leading to its accumulation[10,11]. Likely, neonates also differ in activity of other enzymes, such as uptake into the liver, transporters, and excretion mechanisms, contributing to the bilirubin imbalance, but these factors have only begun to be explored[11].

It is not clear why in some newborns this imbalance is more severe, and clinical presentation worse. Bilirubin accumulation is more pronounced in preterm newborns, reportedly because the relevant metabolic pathways are not yet matured[4], although the specifics of this are not yet established. Some rare pathological factors with clear mechanisms are known, such as hemolysis due to ABO or Rhesus

incompatibility[4] and highly deleterious mutations in *UGT1A1* or transporter genes[8]. Genetic variants may also modify the effects of environmental factors, such as breastfeeding[12]. However, genetic studies of neonatal jaundice to date have been limited to only established candidate genes, and have often produced conflicting results even for the same variants[13]. Most of the genetic basis, and in particular common variants or other, non-UGT genes, remains largely unexplored. Genome-wide association studies (GWAS) could fill in these gaps by discovering new variants or genes associated with neonatal jaundice, as well as clarifying the importance of different known metabolic pathways in newborns, ultimately leading to better identification of risk cases or new treatment strategies.

Here, we performed the first GWAS of neonatal jaundice, using nearly 30,000 parent–offspring trios from Norway. We identified substantial differences from the known associations with bilirubin levels in adults, and provide potential mechanisms for our findings using family structure and eQTL analyses. Our results illustrate that genetic effects can be highly dependent on the context of time and tissue, even in well established metabolic pathways.

## Results

### Missense variant at *UGT1A4* reduces the susceptibility to neonatal jaundice

We conducted a GWAS of neonatal jaundice in 27,384 newborns from the Norwegian Mother, Father and Child Cohort Study (Supplementary Data 1–participants' characteristics, Supplementary Data 2 and Supplementary Fig. 1), with 1826 (6.9%) of them requiring phototherapy treatment, hereafter referred to as cases. We attempted to replicate the identified loci in a meta-analysis of two cohorts from Denmark where case status was defined according to International Classification of Diseases (ICD) codes ($n = 6902$, cases = 1300). Genetic variants at two loci (*UGT1A\** and *CHRDL1*) reached genome-wide significance in the discovery data set ($p$ value $< 5 \times 10^{-8}$, Fig. 1A); none of the two loci harbored secondary, conditionally independent signals reaching genome-wide significance[14]. The polygenicity of neonatal jaundice is relatively modest, with 90% of its heritability condensed in 23.4% of the genome (Supplementary Fig. 2).

Perhaps unsurprisingly, the strongest of the two loci was located in the *UGT1A\** region (2q31.1, Fig. 1B), a complex locus encoding the enzymes responsible for the conjugation of bilirubin, among other substrates. The lead variant (rs17868338) is intronic for several *UGT1A\** genes and in nearly perfect LD ($R^2 = 0.96$) with a missense variant at position 70C > A of *UGT1A4* (rs6755571) that results in the substitution of proline to threonine (p.Pro24Thr). The alternate allele of rs6755571 has a frequency of 6.4% and was associated with a fivefold lower risk of neonatal jaundice (Fig. 1, OR = 0.2; 95% CI = 0.15, 0.25; $p$ value $= 2.7 \times 10^{-55}$). The locus was replicated in a meta-analysis of GWAS of neonatal jaundice in 6902 neonates from two Danish cohorts (cases = 1300), with the same haplotype block leading the association (rs6755571; OR = 0.4; 95% CI = 0.32, 0.50; $p$ value $= 2.4 \times 10^{-15}$, allele frequency = 6.0%; Supplementary Data 2 and Supplementary Fig. 3). We further replicated the association in a cohort of 12,405 neonates (cases = 514) of African American ancestries (rs6755571; OR = 0.50; 95% CI = 0.30, 0.80; $p$ value $= 4.5 \times 10^{-3}$, allele frequency = 1.7%; Supplementary Data 2) and from a cohort of 13,325 neonates (cases = 336) of European American ancestries (rs6755571; OR = 0.46; 95% CI = 0.32, 0.63; $p$ value $= 1.2 \times 10^{-5}$; allele frequency = 5.2%, Supplementary Data 2). In other ancestries, the alternate allele of rs6755571 is even rarer, with a frequency of about 1–2% in South Asians and almost 0% in East Asian samples (dbSNP). For instance, in 54,301 individuals from the Tohoku Medical Megabank Project from Japan, only 1 such allele was observed[15]. We calculated the prevalence of rs6755571 in cases and controls of various ancestries in MoBa and observed effects consistent with several-fold protection across populations (Supplementary Data 3 and Supplementary Fig. 4).

We searched for enrichment in evolutionary metrics around rs6755571 using the GSEL pipeline[16] (Supplementary Data 4). There was an enrichment in haplotype homozygosity metrics, such as cross-population extended haplotype homozygosity (xp-EHH)[17], but the enrichment did not meet statistical significance. Visualization of the extended haplotype homozygosity (EHH) around the missense variant in European and African ancestry showed that the minor allele had a slower decay than the major ancestral allele (Supplementary Fig. 5). The elevated EHH around the minor allele suggests that this region has undergone recent positive selection that has shortened the haplotype around this variant.

Since UGT1A1 is considered to be the only enzyme with substantial glucuronidation activity on bilirubin[18], we further explored the possibility that the association is driven by a regulatory effect on *UGT1A1*. We performed colocalization analysis between neonatal jaundice and *cis*-eQTLs in 127 tissues/cell types from the eQTL Catalog[19], and observed colocalization with *UGT1A1* expression in the colon, but not in the liver (Fig. 1, Supplementary Fig. 6 and Supplementary Data 5). The missense variant was consistently fine-mapped as one of the variants with the largest probability of driving the association with *UGT1A1* expression in the colon and neonatal jaundice (posterior probability > 0.10, Supplementary Fig. 7), with an opposite effect direction (increase of *UGT1A1* expression lowers risk of neonatal jaundice). Importantly, the colocalization between neonatal jaundice and *UGT1A1* gene expression in the colon was observed in two independent eQTL data sets–one derived from tissues from healthy adults (CEDAR), and another based on postmortem tissues (GTEx). The replication of this result, in the same tissue under different conditions increases the robustness of this finding.

The second locus associated with neonatal jaundice is located on the X chromosome, near *CHRDL1* gene (rs12400785, OR = 0.80; 95% CI = 0.77, 0.87; $p$ value $= 3.4 \times 10^{-11}$, MAF = 36.6%). This locus is ~43 Mbp upstream of *G6PD* (its deficiency increases the risk of neonatal jaundice[4,7]), and is likely independent from it. This association was replicated in a meta-analysis of two Danish cohorts (rs12400785; OR = 0.88; 95% CI = 0.80, 0.96; $p$ value = 0.004, $n = 6902$, cases = 1300).

To understand whether the loci identified were driven by the neonate, the mother or both, we evaluated the effects of the parental transmitted and non-transmitted alleles on neonatal jaundice using phased genetic data from 23,196 parent–offspring trios ($n$ cases = 1569). The effect of the missense variant (in the *UGT1A\** region) was limited to the transmitted alleles, indicating a neonate-only effect, independent of the parent-of-origin (Supplementary Fig. 8 and Supplementary Data 2). For the locus on the X chromosome we conducted the analysis separately in girls (maternal transmitted and non-transmitted and paternal transmitted, $n = 11,438$ parent–offspring, $n$ cases = 686) and boys (maternal transmitted and non-transmitted and paternal non-transmitted, $n = 11,758$ parent–offspring, $n$ cases = 883). While in girls, all alleles were borderline associated with neonatal jaundice ($0.018 \leq p$ value $\leq 0.089$), in boys the effect was limited to the maternal transmitted alleles (OR = 0.70, $p$ value $= 3.7 \times 10^{-5}$), suggesting an effect from the parental transmitted alleles. The conditional analysis of the maternal, fetal and paternal dosages supported this, with an effect limited to the neonate (OR = 0.83; 95% CI = 0.76, 0.90; $p$ value $= 1.8 \times 10^{-5}$, Supplementary Fig. 8).

### Maternal GWAS captures the effects of ABO incompatibility on neonatal jaundice

Even though perinatal outcomes are affected by both the fetal and maternal genomes, the investigation of the maternal genome in the etiology of neonatal jaundice so far has been largely overlooked. We performed a maternal GWAS of neonatal jaundice in the offspring ($n = 29,182$, $n$ cases = 2401, Fig. 1A, quantile–quantile plot, Supplementary Fig. 9), identifying three independent loci. One of the signals

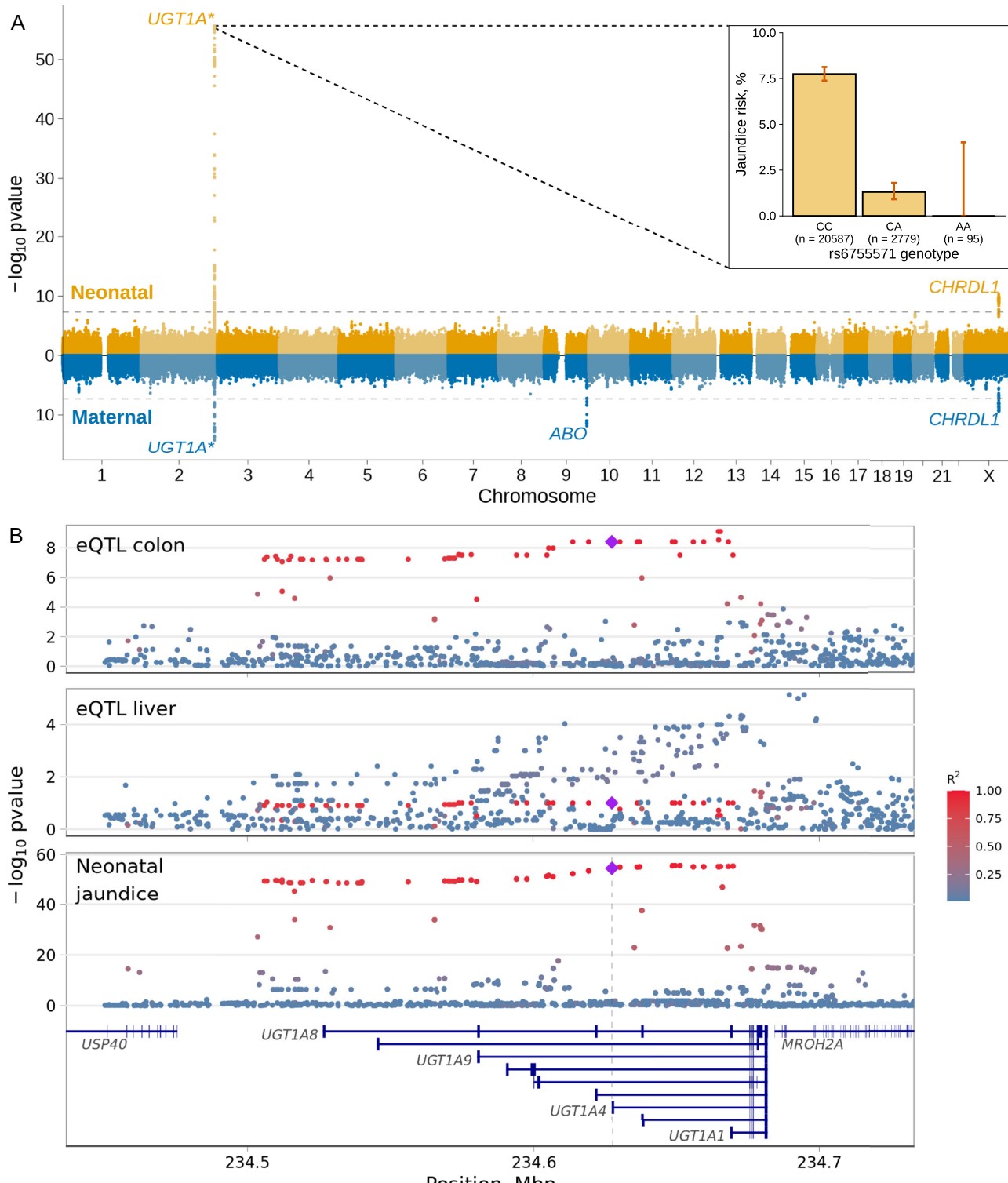

**Fig. 1 | Miami plot of the neonatal and maternal GWAS of neonatal jaundice and associations at *UGT1A\** gene region. A** Miami plot of the GWAS on neonatal jaundice using the neonatal (top, $n = 27{,}384$ neonates, cases = 1826) and maternal (bottom, $n = 29{,}182$, $n$ cases = 2401) genomes. Genome-wide significant loci are named by their nearest protein coding gene or gene family. The $x$-axis shows the chromosome position and the $y$-axis the two-sided $p$ value of the GWAS. The dashed line represents the genome-wide significance threshold ($p$ value = $5 \times 10^{-8}$). The bar plot shows the risk of neonatal jaundice by rs6755571 genotype, a missense variant affecting *UGT1A4*. Error bars in the bar plot represent the 95% CI of the estimate. **B** A regional plot of the genetic associations with *UGT1A1* expression in the colon and liver and with neonatal jaundice at the *UGT1A\** genes region. $Y$-axis shows the two-sided $p$ value of the eQTL associations between the variants and *UGT1A1* expression in colon or liver (from the eQTL Catalog) and neonatal jaundice (from this study, neonatal genome, $n = 27{,}384$, cases = 1826). Highlighted are the missense variant (rs6755571, diamond) and variants in LD with it. LD was estimated in 23,196 non-related neonates from MoBa.

was located at the *UGT1A* * gene region, with the lead SNP (rs17868336) in strong LD with the neonatal missense variant ($R^2 = 0.95$); the association observed with the maternal genome was driven by the parental transmitted alleles (Supplementary Fig. 10), in concordance with a neonate-only effect. A second locus, on chromosome X (Supplementary Data 2), had its lead SNP (rs5942980) in strong LD ($R^2 = 0.95$) with the lead neonatal SNP in the same region. As mentioned above (Supplementary Fig. 8), the effect at this locus was also driven by the transmitted alleles (including the maternal allele), and was thus captured in this GWAS using the maternal genome.

The third locus was located at 9q34.2, in the *ABO* gene region (Fig. 1, rs687621, EAF = 37.2%, OR = 0.46; 95% CI = 0.74, 0.85; $p$ value = $1.4 \times 10^{-12}$). Red blood cell breakdown has a critical role in the development of jaundice in neonates, and is increased in case of maternal–fetal ABO blood group incompatibility. Intrigued by this, we explored the effects of the parental transmitted and non-transmitted alleles on neonatal jaundice for the lead SNP in this locus in 23,196 parent–offspring trios ($n$ cases = 1569). Both maternal alleles (transmitted and non-transmitted) and the paternal transmitted allele of rs687621 were significantly associated with neonatal jaundice, but with opposite effect directions (Supplementary Data 2). To evaluate whether rs687621 reflects maternal–fetal ABO incompatibility, we included genetically derived ABO blood group incompatibility as a covariate (OR = 1.81; 95% CI = 1.57, 2.09; $p$ value = $3.9 \times 10^{-16}$) in the same model as the lead SNP. After adjusting for maternal–fetal ABO incompatibility, the effect estimates of all rs687621 alleles were closer to the null, consistent with its tagging of ABO blood group incompatibility (Fig. 2). It should be noted that the effect of the maternal non-transmitted allele was still significant after conditioning and that this variant is also linked to factor VIII and to blood clotting in general[20].

A GWAS of offspring neonatal jaundice using the paternal genome ($n = 28,384$, $n$ cases = 2361, Supplementary Fig. 11) identified a single locus at the *UGT1A* * genes region (rs149247216), with the lead SNP in strong LD ($R^2 = 0.9$) with the neonatal missense variant (the association was driven by the parental transmitted alleles, Supplementary Fig. 12).

## Missense variant at *UGT1A4* protects from neonatal jaundice despite maternal–fetal ABO incompatibility

As mentioned earlier, bilirubin levels may rise in situations with increased breakdown of red blood cells, such as in maternal–fetal ABO blood group incompatibility. At the same time, the risk of jaundice is exacerbated in neonates born preterm due to immaturity, among others. We sought to investigate whether the protective effects of the missense variant at *UGT1A4* on neonatal jaundice were robust to these risk factors. The effect of rs6755571 on neonatal jaundice was not modified by maternal–fetal ABO blood group incompatibility ($p$ value for interaction = 0.809, Supplementary Fig. 13), but it was stronger during the full term period and post term (linear interaction $p$ value = $6.2 \times 10^{-4}$, Supplementary Fig. 13). This result aligns with a marked postnatal increase in *UGT1A1* expression and protein levels in the small intestine of mice[21].

## Shared and distinct genetic effects on neonatal jaundice and adult bilirubin levels

Finally, we compared the genetic basis of neonatal jaundice with that of adult bilirubin levels at the variant, gene and genome-wide level. The strongest association reported for adult bilirubin levels is also located at the *UGT1A* * region (rs35754645, $p$ value = $3 \times 10^{-26677}$ in GWAS Catalog, proxy variant rs887829, $R^2 = 1$). However, this lead adult SNP was virtually not associated with neonatal jaundice after conditioning for the lead missense variant (OR = 1.2, $p$ value = $8.0 \times 10^{-8}$, Fig. 3; after adjusting for neonatal jaundice lead missense variant rs6755571, OR = 1.12, $p$ value = 0.002), and is also in low LD with our lead variant ($R^2 = 0.03$). In concordance with this, colocalization analysis also

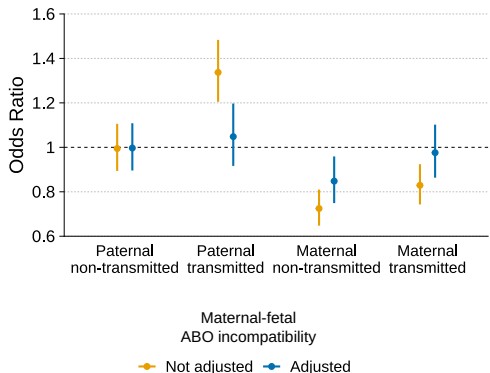

**Fig. 2 | Effects of the parental transmitted and non-transmitted alleles of the *ABO* gene variant (rs687621) on neonatal jaundice before or after adjusting for maternal–fetal ABO blood group incompatibility.** Estimates were obtained from a logistic regression model in 23,196 parent–offspring (cases = 1569) with or without a covariate for genetically determined maternal–fetal ABO blood group incompatibility. The dot represents the odds ratio for the association, and the error bars the 95% CI.

showed that the associations with neonatal jaundice and adult bilirubin levels are driven by distinct variants (posterior probability of sharing the causal variant = $5.2 \times 10^{-24}$, posterior probability of sharing the same locus = 1). Another locus robustly linked with adult bilirubin levels is *SLCO1B1*, a liver-specific transporter that mediates the uptake of bilirubin (top SNP rs4149081, $p$ value = $8 \times 10^{-676}$ in GWAS Catalog). In our GWAS, variants in the *SLCO1B1* gene region were virtually not associated with neonatal jaundice (rs4149081 OR = 1.11, $p$ value = 0.02, Supplementary Fig. 14).

Next, we assessed whether the differences are also present at the genome-wide scale by evaluating the effects of a polygenic score of adult bilirubin levels on neonatal jaundice risk, with or without excluding chromosome 2, which harbors the *UGT1A* * genes region. To generate such scores, we extracted weights from a previously published score that was trained and validated in adults from the UK Biobank[22]. The polygenic score of adult bilirubin levels had a bimodal distribution, but only when including chromosome 2 (Supplementary Fig. 15). As could be anticipated, the polygenic score of adult bilirubin levels was strongly associated with neonatal jaundice risk (Fig. 3, $n = 23,196$, cases = 1569, OR = 1.18/standard deviation; 95% CI = 1.12, 1.23; $p$ value = $1.4 \times 10^{-10}$). However, once we exclude chromosome two from the polygenic score, the effect of the score was close to null (OR = 1.05/standard deviation; 95% CI = 0.99, 1.11; $p$ value = 0.064).

Finally, we conducted phenome-wide colocalization analyses between neonatal jaundice and adult bilirubin levels and 952 phenotypes from the PAN UK Biobank (using only summary statistics derived from individuals of European ancestries) at the UGT genes region (Supplementary Fig. 16 and Supplementary Data 6). Overall, we observed that only a small number of phenotypes colocalized with either neonatal jaundice or adult bilirubin levels ($n = 5$, 6 for neonatal jaundice and adult bilirubin levels, respectively). Total cholesterol levels colocalized with both neonatal jaundice and adult bilirubin levels, due to two conditionally independent lead SNPs associated with total cholesterol levels in the region. This result is in line with the negative correlation observed between bilirubin levels and lower total cholesterol levels, but this association was not replicated in a Mendelian randomization analysis[23]. We further observed high probability of colocalization (posterior probability > 0.9) between neonatal jaundice and a number of lipid-related traits, and between adult bilirubin levels and phenotypes related to gallstones (i.e. cholelithiasis), which can be caused by excess of bilirubin levels.

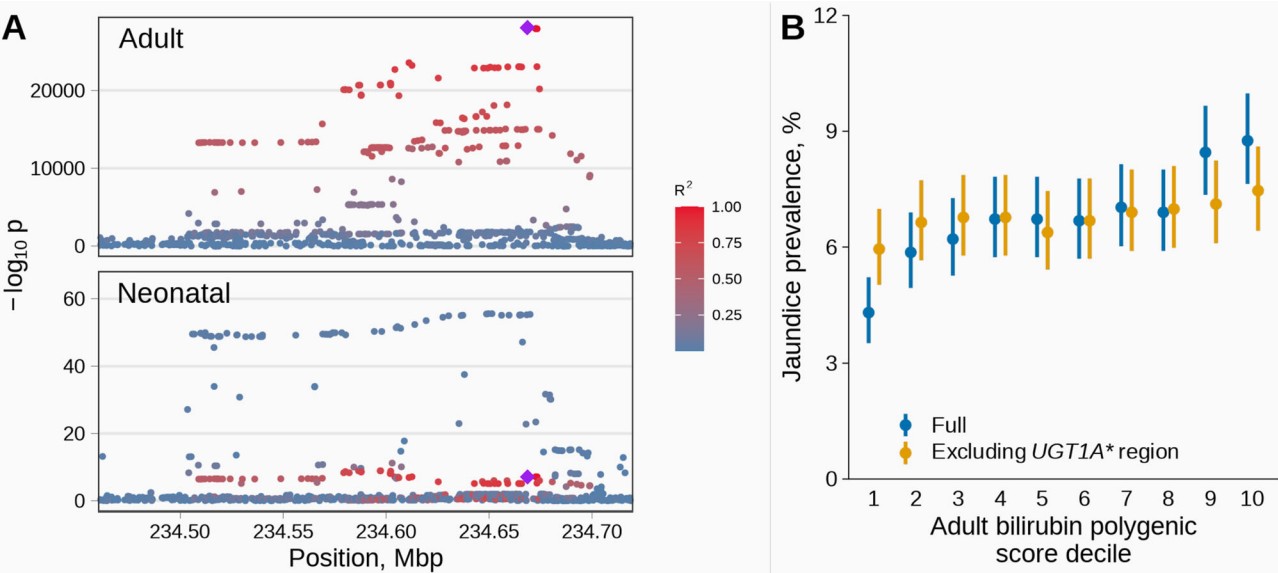

**Fig. 3 | Neonatal jaundice effects of adult bilirubin level variants and polygenic score. A** Genetic effects of the variants at the *UGT1A\** genes region. Y-axis shows the two-sided *p* value of the associations between the variants and adult bilirubin levels (a previous publication) or neonatal jaundice (from this study, neonatal genome, *n* = 27,384, cases = 1826). Highlighted are the top adult variant (rs887829, diamond) and variants in LD with it. LD was estimated in 23,196 non-related mothers from MoBa. **B** Jaundice prevalence by polygenic score of adult bilirubin levels, including or excluding the *UGT1A\** genes region, observed in 23,196 neonates (cases = 1569). Weights for the polygenic score of adult bilirubin levels were obtained from the PGS Catalog (ID: PGS002160). Dots represent the neonatal jaundice prevalence, and the error bars the 95% CI.

## Discussion

Bilirubin levels in adults are known to have a strong heritable component, but genetic studies of neonatal jaundice so far have been limited. We conducted a GWAS of neonatal jaundice using the maternal, neonatal and paternal genomes in nearly 30,000 families from Norway. The maternal GWAS of offspring neonatal jaundice recapitulated known biology at the *ABO* gene region, where the association of the parental transmitted and non-transmitted alleles was explained by maternal–fetal blood group incompatibility. However, we also found notable differences between the adult and neonatal genetics of jaundice—both at the variant and gene levels—which may indicate an unknown, context-specific mechanism or regulatory aspect of bilirubin clearance in neonates.

The most striking result is the *UGT1A4* missense variant with a fivefold effect on the risk of neonatal jaundice (rs6755571)—the protective effect was replicated in cohorts of Nordic, African American and European American ancestries. Such an effect size is very unusual in common variants, and is closer to those seen in rare loss-of-function mutations; together with its position, it is thus tempting to speculate that the alternate, protective allele causes a substantial activity change in the UGT1A4 enzyme. However, unlike UGT1A1, wild-type UGT1A4 shows very little bilirubin conjugation activity[18]. Second, the variant is located in the signal peptide of UGT1A4, which is removed from the mature protein; Troberg and Finel showed that after cleavage the protein alleles only differ by two additional amino acids in the N terminus[24]. The mutated protein has lower activity on some substrates, although in human cell extracts only small differences were observed, possibly due to efficient removal of the mis-folded proteins[24]. In light of this, we consider it unlikely that the variant introduces bilirubin conjugation activity in this protein, and so probably does not act through UGT1A4 directly.

More likely, the top associated variant, or one or more variants in close LD with it, still act through UGT1A1, the primary bilirubin conjugation enzyme. Our eQTL colocalization result suggests that this could be through regulation of *UGT1A1* expression in the intestines. Intestinal *UGT1A1* expression is known in adult humans, although overshadowed by liver bilirubin metabolism[18,25]; in human neonates it has not been investigated. Our associations suggest that intestinal UGT1A1 could be the primary factor determining bilirubin accumulation in human neonates. In mice with transgenic *UGT1A\**, Fujiwara et al.[21] observed that during the neonatal period the *UGT1A1* gene is in fact not expressed in the liver, and only in the intestines. Furthermore, the intestine expression level was not affected by the *UGT1A1\*28* variant commonly associated with adult hyperbilirubinemia[21], which is consistent with us not replicating the adult GWAS hits. (The *\*28* variant is in LD with the top hits for adult bilirubin levels[26].) Another interesting possibility is that the causal variant is still the missense variant affecting *UGT1A4*, and it acts by modifying the activity of mature UGT1A1—studies have suggested that these proteins can form a heterodimer, and their relative levels affect their catalytic properties[27]. In either case, our results point toward marked differences from the standard etiology of hyperbilirubinemia in adults.

Outside the *UGT1A\** locus, another difference from adults that we observed is the lack of association at *SLCO1B1*. This gene encodes a protein responsible for transport of conjugated bilirubin between nearby hepatocytes, and notably has low expression in neonates[11]. Thus it seems that the rate of such transport is not the bottleneck in bilirubin clearance in neonates, possibly due to low activity of UGT1A1, or again due to some amount of conjugation occurring outside the liver. Finally, the additional association that we observe at the *CHRDL1* locus does not immediately relate to any known mechanism of jaundice.

Despite the large effect sizes and notable functional consequences of our reported variants, we can only infer limited details about the mechanisms behind these associations. The differences observed in the genetic control of a single pathway—with adult bilirubin level SNPs showing some of the lowest *p* values ever observed in a GWAS and not detected for neonates—show remarkable context-specific mechanisms. To make further progress in post-GWAS inference, both for this and other traits, expression data covering a wide range of tissues and conditions, finely resolved in time, and highly detailed in terms of protein variants will be needed. There are also

genetic questions that remain open. It is unclear why the alleles observed here with large protective neonatal effects—thus presumably highly selected for—are relatively uncommon, and almost absent in some populations. More broadly, the prevalence of missense variants with regulatory effects on genes distinct from the one it encodes is not well established and, if common, may introduce artifacts in variant-to-gene mapping efforts.

In light of the many, often conflicting candidate gene studies of neonatal jaundice, this study provides a solid basis for further investigations into the etiology of this trait and a marked departure from the genetics of adult bilirubin metabolism.

# Methods

## Ethics
All study participants provided a signed informed consent, and the study protocol has been approved by the administrative board of the Norwegian Mother, Father and Child Cohort Study, led by the Norwegian Institute of Public Health. The establishment of MoBa and initial data collection was based on a license from the Norwegian Data Protection Agency and approval from The Regional Committee for Medical Research Ethics. The study was approved by the Norwegian Regional Committee for Medical and Health Research Ethics South-East (2015/2425) and by the Swedish Ethical Review Authority (Dnr 2022-03248-01).

The DNBC mothers provided written informed consent on behalf of themselves and their children. The study protocol was approved by the Regional Scientific Ethical Committee of Copenhagen and the Danish Data Protection Agency. For the Statens Serum Institut's genetic epidemiology (SSI-GE) cohorts, GWAS data were generated based on dried blood spot samples obtained during routine neonatal screening and stored in the Danish Neonatal Screening Biobank, which is part of the Danish National Biobank. Parents are informed in writing about the neonatal screening and that the samples can later be used for research, pending approval from relevant authorities. The SSI-GE studies were approved by the Regional Scientific Ethical Committee of Copenhagen, the Danish Data Protection Agency and the Danish Neonatal Screening Biobank Steering Committee.

## Study population
The Norwegian Mother, Father and Child Cohort Study (MoBa) is a family-based cohort study that enrolled 114,000 children, 95,000 mothers and 75,000 fathers from 50 hospitals across Norway[28]. Here, we used a subset of MoBa participants that were genotyped, comprising a total of 98,109 samples (~25,000 parent–offspring trios). Information regarding maternal and neonatal health and complications of pregnancy and birth were extracted from the Medical Birth Registry of Norway and linked to questionnaires (v12 of the quality-assured data files) and genetic data generated in MoBa.

Most of the samples chosen for genotyping, and hence included in the present analysis, fulfilled the following requirements: singleton pregnancy, live-birth, data linked to the Norwegian Medical Birth Registry, birth weight and length available, the mother had answered at least the first questionnaire (at 15–17 gestational weeks), and parental DNA samples were available. To ensure a minimal concordance between different genotyping batches, we limited this study to the analysis of live-born singleton neonates who did not die in their 1st year of life, and their parents.

As mentioned earlier, neonatal jaundice is a matter of degree. In Norway, all newborns are assessed for neonatal jaundice by clinicians, and cases are routinely reported in a checkbox in the Medical Birth Registry of Norway. The definition only includes neonates requiring jaundice treatment (phototherapy or exchange transfusion) according to national consensus guidelines[3].

## Genotyping and quality control
Genotyping was performed using four different arrays: 33,538 samples with Illumina Human Core Exome (Genomics Core Facility, Norwegian University of Science and Technology, Trondheim, Norway), 26,990 with Illumina Infinium Global Screening Array (GSA) MD (Erasmus Medical Center, Erasmus University, Rotterdam, The Netherlands), 5959 with Illumina Human Omni Express (deCode Genetics, Reykjavik, Iceland), 17,730 with Illumina Infinium Omni Express (deCode Genetics, Reykjavik, Iceland) and 9632 samples with Infinium GSA 24 (deCode Genetics, Reykjavik, Iceland). Genome coordinates were mapped to the Genome Reference Consortium Human Build 37 (hg19). Genotypes were called in Illumina GenomeStudio software (v.2011.1 and v.2.0.3). Samples with call rate <0.98, excess heterozygosity >4 SD or sex mismatch, and variants with call rates <98%, 10% GenCall-score <0.3, cluster separation <0.4, Theta AA standard deviation >0.4 and Hardy–Weinberg equilibrium test $p$ value < $10^{-6}$ were excluded. Finally, individuals of primarily non-European ancestries were excluded based on principal components with reference samples from the HapMap project, version 3.

## Phasing and imputation
Phasing was conducted locally using Shapeit v2.790[29], including mother–child pairs whenever available using --duohmm method. Prior to imputation, insertions and deletions were removed and allele, marker position and strand were updated to match the Haplotype Reference Consortium (HRC) imputation panel (v1.1)[30]. Imputation was performed at the Sanger Imputation Server with positional Burrows–Wheeler transform[31] and HRC imputation panel (v1.1).

## Genome-wide association study
We estimated the associations between neonatal ($n = 27,384$, $n$ cases = 1826), maternal ($n = 29,182$, $n$ cases = 2401) and paternal ($n = 28,384$, $n$ cases = 2361) genetic dosages on neonatal jaundice using logistic regression with Firth correction implemented in REGENIE (v3.1)[32]. The analysis was adjusted for genotype principal components and sex, and was based on SNPs with an imputation INFO score > 0.7 and minor allele frequency ≥ 0.1%. Subsequent analyses were based on unrelated parent–offspring triads ($n = 23,196$, total samples = 68,361, kinship coefficient ≤ 0.125). Genomic inflation factors for all three GWAS were very close to one, and so we did not adjust $p$ values for them.

## Replication analyses
**Danish cohorts.** We attempted to replicate the loci identified in the neonatal GWAS of jaundice in a meta-analysis of two Danish cohorts. The individuals included in the analyses were participants in the Danish National Birth Cohort (DNBC) and SSI-GE studies, who have existing genotype data from previous studies available. SSI-GE consists of samples from studies of febrile seizures, atrial septal defects, Hirschsprung disease, opioid dependence, hydrocephalus, post-partum depression and hypospadias. Phenotypic information to define patients with neonatal jaundice was retrieved from the Danish National Patient Register[33]. The register includes information from all inpatient admissions in Denmark since 1977 and all emergency and outpatient hospital contacts since 1995. Diagnoses are coded according to the ICD using the 8th revision (ICD-8) from 1977 to 1993 and the 10th revision (ICD-10) from 1994 onwards. Eligible neonatal jaundice cases were defined as individuals with ICD-8 code 77891 or ICD-10 codes DP58 and DP59 in the register. Information about gestational duration was retrieved from the Danish Medical Birth Register[34].

Genotyping, data cleaning and imputation of the DNBC and SSI-GE individuals were done as previously described[35]. The HRC release 1.1 panel was used for imputation, and 7.5 million variants with minor allele frequency larger than 1% and imputation INFO > 0.8 were included in the analyses. Individuals of non-European ancestries (outliers in

principal component analysis) were excluded prior to analysis. In the DNBC sample, 719 cases and 2994 controls were included. In the SSI-GE sample, we balanced the ratio of neonatal jaundice cases to controls, so that it was the same within each of the disease groups of the original studies. In total, 581 cases and 2608 controls were included from the SSI-GE studies.

Association analyses were done under an additive genetic model including sex, gestational duration and five principal components as covariates using REGENIE 2.2.3[32]. Results from DNBC and SSI-GE were combined using fixed-effects inverse-variance-weighted meta-analysis as implemented in METAL[36].

**African American and European American cohorts.** The study initially included two distinct groups based on genotyping platforms. One group comprised 24,915 children whose genetic profiles were determined using the HumanHap 550 + 610 series of Illumina SNP chips. The second group consisted of 16,954 children genotyped via the GSA series of Illumina SNP chips. To ensure the precision of the analysis, the samples were categorized according to ancestry. Subsequent analyses were tailored to these delineated sub-cohorts, taking into account both their ancestral backgrounds and the specific genotyping chip employed.

Quality control of the genetic data was performed using PLINK 2.0[37]. The following criteria were applied for quality control: exclusion of genetic variants with a minor allele frequency <1%, failing the Hardy–Weinberg Equilibrium test ($p < 1 \times 10^{-6}$), and with a genotyping rate <95%. Additionally, individuals with a genotype call rate of <95% were excluded. Relatedness checks were also conducted, removing any duplicates or first-degree relatives. Imputation of the genotype data was performed using the TOPMed Imputation Server[38]. The reference panel consisted of a diverse set of individuals, providing high-quality imputation across various ethnic groups

The association analysis between SNPs and the trait of interest was conducted using SNPTEST v2.5[39]. The analysis was adjusted for the top principal components from the PCA, along with child sex. Additive models were tested using the score method.

We conducted a fixed-effects meta-analysis of the data obtained from both the HumanHap 550 and GSA chip sets using the genome-wide association meta-analysis (GWAMA) software[40]. Following stringent quality control measures, the final composition of the study cohorts was established as follows:

- African Ancestry (HumanHap 550): 311 individuals were identified as cases (170 males and 141 females), alongside 6608 controls (3237 males and 3371 females).
- African Ancestry (GSA): the case group included 203 individuals (103 males and 100 females), with a control group comprising 5283 individuals (2439 males and 2844 females).
- European Ancestry (HumanHap 550): the final case tally was 207 individuals (138 males and 69 females), with 7706 controls (4052 males and 3654 females).
- European Ancestry (GSA): there were 129 cases (62 males and 67 females), and the control group included 7817 individuals (3893 males and 3924 females).

**Non-European ancestry analysis in the Norwegian Mother, Father and Child Cohort Study.** We inspected the effects of the top hits in samples not limited to European ancestry. For this, samples from 1000 Genomes Project[41] were projected to the same principal components as the MoBa data (see genotyping and quality control). Ancestry of the MoBa children was then inferred using a nearest centroid classifier with the first three PCs as features; i.e. the centroid of each 1000 Genomes population was calculated, and samples assigned to the population with the closest centroid. We report allele frequencies by inferred population and case/control status.

**Evolutionary analyses**
The evolutionary context of the SNP rs6755571 was examined using several methods. First, we used the GSEL pipeline[16] to test for enrichment across 13 different evolutionary metrics. This method compares the evolutionary metrics for a region of the genome to control regions extracted to match minor allele frequency and LD patterns. We conducted this analysis using the variants in high LD ($r^2 > 0.9$) within 500 kb of rs6755571. These linked SNPs were identified using the web Ensembl BioMart for the hg19 version of the human genome[42]. This produced 785 variants in high LD. This enables the program to identify evolutionary patterns that may impact genomic regions as opposed to singular SNPs. We also lowered the number of matched control sets to 200 due to the low minor allele frequency of rs6755571, which made it difficult to find other representative control SNPs. We further investigated the haplotype homozygosity of the region using the rehh 3.2.2 package in R[43]. We ran this analysis using the 1000 Genomes project phase 3 data divided into European, African and East Asian ancestries. We calculated and visualized the decay of haplotype homozygosity around the focal SNP using this package. The iHS value for the variant was computed using all variants on chromosome 2 for binning.

**Conditional analysis**
We looked for secondary, conditionally independent associations within each locus using approximate Conditional and Joint (COJO) analysis[14] implemented in the GCTA software[44]. To identify such variants ($p$ value $< 5 \times 10^{-8}$), we ran a stepwise model selection (-cojo-slct) for each of the genome-wide significant loci (±1.5 Mb from the lead SNP). LD between genetic variants was estimated from the same samples with which the GWAS were run, but excluding samples that were genetically related. To complement this analysis, we performed conditional analysis using individual level genetic data by adding the leading fetal variant in the *UGT1A\** genes region to the models.

**Local SNP heritability**
We estimated local SNP heritability using Heritability Estimation from Summary Statistics (v.0.5.3)[45]. This method estimates narrow sense heritability using GWAS summary statistics and a reference sample for LD (1000 Genomes phase 3 SNPs with MAF > 0.05) in approximately LD-independent regions[46]. As a measure of polygenicity, we ranked all genomic loci by the proportion of heritability they explained and calculated the proportion of the genome occupied by each region. We visually contrasted its polygenicity against that of height, using summary statistics from a previously published GWAS[47].

**Resolving maternal–fetal effect origin**
We classified the effects of the lead SNPs as having only maternal, only fetal, or maternal and fetal effects using the parental transmitted and non-transmitted alleles. By using phased genotype data (i.e. estimated haplotypes) in triads, we inferred the parent-of-origin of the alleles in the offspring, as previously described[48,49]. Once the transmitted alleles were allocated to a parent, the non-transmitted maternal and paternal alleles were extracted. As mentioned above, genetic data were phased with SHAPEIT2[29], which uses a hidden Markov model in combination with pedigree information to refine phase calls and assign parents to haplotypes.

For each lead SNP we fit the following logistic regression model in 23,196 parent–offspring triads ($n$ cases = 1569):

$$\text{Neonatal jaundice} = MnT + MT + PT + PnT + \text{PCs} + \text{Fetal sex} + \text{Batch}$$

(1)

where $MnT$ and $MT$ refer to the maternal non-transmitted and transmitted alleles, respectively, and $PnT$ and $PT$ refer to the paternal non-transmitted and transmitted alleles, respectively. In addition to

teasing out maternal–fetal effects, this analysis allowed us to assess whether the effects observed in the fetus were limited to one of the parental alleles or were independent of the parent-of-origin. Analyses on chromosome X were conducted in girls (maternal transmitted and non-transmitted and paternal transmitted, $n = 11{,}438$ parent–offspring, $n$ cases = 686) and boys (maternal transmitted and non-transmitted and paternal non-transmitted, $n = 11{,}758$ parent–offspring, $n$ cases = 883) separately. Due to the lower sample size of this analysis, we also performed conditional analysis using maternal, paternal and neonatal dosages as follows:

$$\text{Neonatal jaundice} = \text{Maternal} + \text{Neonatal} + \text{Paternal} + \text{PCs} + \text{Fetal sex} + \text{Batch} \quad (2)$$

### Genetic determination of ABO blood group
We determined the ABO blood group of mothers and their offspring based on two SNPs (rs8176746 and rs8176719) located in the ABO gene, as previously shown[50]. The reference panel we used for imputation does not include indels (rs8176719), so we chose rs657152 instead, which is in close proximity and high LD with rs8176719 ($R^2 = 0.98$). The rs8176746 missense variant defines the B blood group and the rs8176719 defines the O blood group (Supplementary Data 7 and 8). We defined maternal–fetal ABO blood group incompatibility as when the mother's blood group was O, and that of the neonate was A or B; all other maternal–fetal blood group combinations were treated as compatible.

### Colocalization analysis with expression QTLs
To assess potential regulatory effects of the neonatal locus at the *UGT1A\** genes region, we conducted colocalization analyses with *cis*-eQTLs (1 Mb around gene transcription start site) from the eQTL Catalog[16]. We limited the analysis to eQTLs for only seven of the *UGT1A\** genes (*UGT1A1, UGT1A4, UGT1A6, UGT1A7, UGT1A8, UGT1A9* and *UGT1A10*) and to gene expression data from either RNA-sequencing or microarrays studies, totaling 127 different cell types/tissues. To match the genome reference for our data and that from the eQTL Catalog, we lifted over our GWAS summary statistics (originally mapped to GRCh37) to the Genome Reference Consortium Human Build 38 by using the UCSC liftOver tool[51]. Colocalization analyses were performed using coloc package for R[52]. In a Bayesian statistical framework, this analysis assesses whether the associations with two traits within a single genomic region are explained by shared or distinct variants in close LD, assuming that only one causal variant exists for the two traits in the specified region. We used conservative priors than the ones set by default (prior for shared causal variant = $5 \times 10^{-6}$). While we have tried to convey the information without setting a harsh threshold for colocalization, we defined strong evidence of colocalization whenever the posterior probability of colocalization was >0.9.

### Comparison with adult bilirubin levels
We downloaded summary statistics from the largest GWAS of adult bilirubin levels[53] performed in individuals from predominantly principal component-derived European ancestries. We conducted colocalization analysis in the region spanning *UGT1A\** genes between neonatal jaundice and adult bilirubin levels. At the variant level, we inspected the associations between variants and neonatal jaundice and adult bilirubin levels in the UGT1A\* and *SLCO1B1* genes region, and visualized them by plotting the associations. For the *UGT1A\** genes region, LD between variants was obtained from non-related samples of mothers used as part of this GWAS ($n = 23{,}196$), and for variants with the *SLCO1B1*, the online tool LocusZoom[54] was employed, with LD estimated in 1000 Genomes Project (EUR superpopulation). Lead SNPs for adult bilirubin levels were obtained from the GWAS Catalog[55].

At the genome-wide scale, we evaluated the effect of a polygenic score of adult bilirubin levels on neonatal jaundice with or without including the *UGT1A\** genes region in the score (defined as ±1 Mb around the missense variant−rs6755571). To construct the polygenic score, we obtained weights from a previously published score trained and validated in adults from the UK Biobank[22] and deposited in the PGS Catalog (ID: PGS002160)[56].

We finally performed colocalization analyses between neonatal jaundice and adult bilirubin levels and 952 phenotypes from the PAN UK Biobank, using summary statistics derived from individuals of European ancestry. We did not pre-select traits, and thus used all summary statistics where the trait had a significant heritability estimate above 1%, and passed the quality control metrics suggested by the PAN UK Biobank Team. Colocalization analyses were conducted as for the eQTL colocalization analysis mentioned above.

### Reporting summary
Further information on research design is available in the Nature Portfolio Reporting Summary linked to this article.

## Data availability
Summary statistics from the three GWAS of neonatal jaundice can be accessed at https://www.fhi.no/en/studies/moba/for-forskere-artikler/gwas-data-from-moba/. Data from the Norwegian Mother, Father and Child Cohort Study and the Medical Birth Registry of Norway used in this study are managed by the national health register holders in Norway (Norwegian Institute of Public Health) and can be made available to researchers, provided approval from the Regional Committees for Medical and Health Research Ethics (REC), compliance with the EU General Data Protection Regulation (GDPR) and approval from the data owners. The consent given by the participants does not open for storage of data on an individual level in repositories or journals. Researchers who want access to data sets should apply through helsedata.no. Access to data sets requires approval from The Regional Committee for Medical and Health Research Ethics in Norway and an agreement with MoBa. Summary statistics from adult bilirubin levels can be accessed from https://doi.org/10.35092/yhjc.12355382. Data from the PAN UK Biobank can be accessed from https://pan.ukbb.broadinstitute.org/.

## Code availability
All code for this project is available at https://github.com/PerinatalLab/neo-jaundice.

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

## Acknowledgements

The Norwegian Mother, Father and Child Cohort Study is supported by the Norwegian Ministry of Health and Care Services and the Ministry of Education and Research. We are grateful to all the participating families in Norway who take part in this on-going cohort study. We thank the Norwegian Institute of Public Health (NIPH) for generating high-quality genomic data. This research is part of the HARVEST collaboration, supported by the Research Council of Norway (#229624). We also thank the NORMENT Centre for providing genotype data, funded by the Research Council of Norway (#223273), South East Norway Health Authorities and Stiftelsen Kristian Gerhard Jebsen. We further thank the Center for Diabetes Research, the University of Bergen for providing genotype data and performing quality control and imputation of the data funded by the ERC AdG project SELECTionPREDISPOSED, Stiftelsen Kristian Gerhard Jebsen, Trond Mohn Foundation, the Research Council of Norway, the Novo Nordisk Foundation, the University of Bergen and the Western Norway Health Authorities. B.J. received funding from The Swedish Research Council, Stockholm, Sweden (2019-01004), The Research Council of Norway, Oslo, Norway (FRIMEDBIO #547711), March of Dimes (#21-FY16-121), Agreement concerning research and education of doctors (ALFGBG-965353). This study was in part supported by the Eunice Kennedy Shriver National Institute Of Child Health & Human Development of the National Institutes of Health under Award Number R01HD101669. The content is solely the responsibility of the authors and does not necessarily represent the official views of the National Institutes of Health. G.Z. is supported by the Burroughs Wellcome Fund (Grant 10172896), and the March of Dimes Prematurity Research Center Ohio Collaborative. P.S.N received funding from Längmanska Kulturfonden. B.F. was partially supported by a grant from the Novo Nordisk Foundation (NNF17OC0027594). We thank the UK Biobank participants and the PAN UK Biobank Team for facilitating access to summary statistics for GWAS of as many phenotypes in as many diverse populations as possible.

## Author contributions

P.S.N., J.J. and B.J. conceived and conceptualized the study. P.S.N., J.J., K.Y., X.W., A.L.L. and J.P.B. conducted formal analysis. P.S.N. and J.J. wrote the original draft. B.F., A.L.L. and J.P.B. participated in writing the methods. J.J., M.V., O.H., C.F. and F.G. performed data curation and quality control. P.M., A.L.L., M.F., M.Z., P.L., O.A., P.N., B.F., L.J.M., S.J., G.Z., S.F.A.G., H.H. and B.J. provided critical resources. P.S.N., B.J., G.Z. and S.J. supervised the project. All authors reviewed, edited and approved the final version of the manuscript.

## Funding

## Competing interests

The authors declare no competing interests.

## Additional information

[1]Department of Obstetrics and Gynaecology, Sahlgrenska Academy, Institute of Clinical Sciences, University of Gothenburg, Gothenburg, Sweden. [2]Department of Epidemiology Research, Statens Serum Institut, Copenhagen, Denmark. [3]Copenhagen University Hospital Biobank Unit, Department of Clinical Immunology, Rigshospitalet, Copenhagen, Denmark. [4]Center for Applied Genomics, Children's Hospital of Philadelphia, Philadelphia, PA, USA. [5]Quantinuum Research LLC, Wayne, PA, USA. [6]Mohn Center for Diabetes Precision Medicine, Department of Clinical Science, University of Bergen, Bergen, Norway. [7]Department of Genetics and Bioinformatics, Health Data and Digitalization, Norwegian Institute of Public Health, Oslo, Norway. [8]Department of Bioinformatics and Genomics, College of Computing and Informatics, North Carolina Research Campus, University of North Carolina at Charlotte, Kannapolis, NC, USA. [9]Division of Pharmaceutical Chemistry and Technology, Faculty of Pharmacy, University of Helsinki, Helsinki, Finland. [10]Department of Pharmaceutical Sciences, College of Pharmacy and Pharmaceutical Sciences, Washington State University, Spokane, WA, USA. [11]Division of Human Genetics, Children's Hospital of Philadelphia, Philadelphia, PA, USA. [12]Department of Pediatrics, Perelman School of Medicine, University of Pennsylvania, Philadelphia, PA, USA. [13]Division of Pulmonary Medicine, Children's Hospital of Philadelphia, Philadelphia, PA, USA. [14]Centre for Fertility and Health, Norwegian Institute of Public Health, Oslo, Norway. [15]NORMENT Centre, University of Oslo, Oslo, Norway. [16]Institute of Clinical Medicine, Faculty of Medicine, University of Oslo, Oslo, Norway. [17]Division of Mental Health and Addiction, Oslo University Hospital, Oslo, Norway. [18]Children and Youth Clinic, Haukeland University Hospital,

Bergen, Norway. [19]Department of Genetics, Perelman School of Medicine, University of Pennsylvania, Philadelphia, PA, USA. [20]Division of Endocrinology and Diabetes, Children's Hospital of Philadelphia, Philadelphia, PA, USA. [21]Department of Biology, University of Copenhagen, Copenhagen, Denmark. [22]Office of the President, Burroughs Wellcome Fund, Research Triangle Park, NC, USA. [23]Division of Human Genetics, Center for the Prevention of Preterm Birth, Perinatal Institute, Cincinnati Children's Hospital Medical Center, Cincinnati, OH, USA. [24]Department of Pediatrics, University of Cincinnati College of Medicine, Cincinnati, OH, USA. [25]Department of Medical Genetics, Haukeland University Hospital, Bergen, Norway. [26]Present address: Department of Pharmaceutical Sciences, School of Pharmacy and Pharmaceutical Sciences, University at Buffalo, Buffalo, NY, USA. [27]These authors contributed equally: Pol Solé-Navais, Julius Juodakis. ✉e-mail: pol.sole.navais@gu.se; bo.jacobsson@obgyn.gu.se

