## [Peer Review File · Nature Communications]

REVIEWER COMMENTS

Reviewer #1 (Remarks to the Author):

This is a novel GWAS of neonatal jaundice, a clinically relevant issue, based on a discovery cohort of almost 30,000 neonate/mother/father trios from the Norwegian Mother Father and Child Cohort Study (MoBA) cohort, with the following findings:

1. Identification of two genome-wide significant loci (UGT1A in chromosome 2, CHRDL1 in chromosome X) that were further replicated in external Danish cohorts. While the lead variant for UGT1A was intronic, the alternate allele of a common missense variant at UGT1A4 (with a proline to threonine substitution, in high LD with the intronic lead variant and with reduced bilirubin binding) was associated with ~5-fold reduced risk for neonatal jaundice.
2. Further eQTL colocalization analyses showed association with UGT1A1 expression in the colon but not the liver—supporting a role for intestinal (rather than hepatic) UGT1A1 in neonatal jaundice.
3. Parental GWASs identified a third locus in ABO gene region in the maternal genome, consistent with a role for ABO incompatibility in neonatal jaundice. Further studies showed the relevance of transmitted (rather than non-transmitted) parental alleles on neonatal jaundice.
4. Genetic variants associated with adult jaundice were not associated with those identified in neonatal jaundice. However, PRS of adult bilirubin levels were strongly associated with neonatal jaundice risk, although this effect was driven by UGT1A region in chromosome 2.

Strengths of this study is the novelty in conducting a GWAS of this clinically relevant issue of neonatal jaundice using a large, unique cohort of neonate/mother/father trio—with findings that were further replicated with potential physiological implications in intestinal vs hepatic expression, even though the findings related to UGT1A or ABO are not unexpected. Some limitations include the lack of diversity in using cohorts of Nordic European ancestry and absence of specific mechanistic or functional analyses, as the authors are aware. Could include a phenome-wide association study for the variants identified at least.

Reviewer #2 (Remarks to the Author):

The authors have carried out a genome-wide association study (GWAS) of neonatal jaundice on a sample of 30,000 parents-offspring trios from Norway). Cases were approx. 2000. Although the authors conclude

that there are marked differences between the pathways involved in neonatal jaundice compared to those regulating bilirubin levels in adults, suggesting a distinct biological basis, these conclusions are not well based on the results. Similarly, the methodology used in the study to verify the proposed objective is not the correct one.

It is known that elevated bilirubin concentrations, primarily those genetically determined from childhood, are due to genetic variants in the UGT complex genes on chromosome 2. Specific polymorphisms can be very diverse depending on the geographical origin of the population studied. In this case it is a population from Northern Europe and the prevalence of genetic variants in these UGT1A* genes may be very different from the variants that can be found in the population of Southern Europe. There may still be more differences in genetic variants in UGT genes if non-European populations are analyzed.

The protective variant highlighted by the authors as protective (rs6755571, A allele OR = 0.2, value = 2.7×10^{-55} , frequency = 6.4%, may be a variant with a great founder effect in such a population, so additional GWAS in diverse populations (Southern Europe, Hispanic, Asian, African, etc.) are needed to bring more knowledge about the genetics of bilirubin concentrations in children and adults and support the author's conclusions. It is also known that these additional genes not located on chromosome 2 may be different depending on the population analysed. The authors would therefore need to analyze more diverse adult populations to better quantify the most relevant genes in each geographical origin and not just focus on populations in Northern Europe. With this analysis they could verify that the genetic risk scores (basically the non-UGT genes) are heterogeneous in the adult population and can in turn be modulated by lifestyle variables.

Similarly, in the study of children, the authors may have analyzed genetic modulations through other factors of parental style and in pregnancy to provide new information on new interactions with genetic variants in that population.

Another limitation is that the authors perform functional analyses in silico. This implies limitations on data sources and conclusions. It would have been more appropriate to do some expression analysis in the wet lab.

In general, the article has significant limitations and more analyses are needed in other diverse populations and with better methodology.

REVIEWER COMMENTS

We thank the editor and reviewers for the positive assessment of our work and highly appreciate the comments and suggestions.

Reviewer #1 (Remarks to the Author):

This is a novel GWAS of neonatal jaundice, a clinically relevant issue, based on a discovery cohort of almost 30,000 neonate/mother/father trios from the Norwegian Mother Father and Child Cohort Study (MoBA) cohort, with the following findings:

1. Identification of two genome-wide significant loci (UGT1A in chromosome 2, CHRDL1 in chromosome X) that were further replicated in external Danish cohorts. While the lead variant for UGT1A was intronic, the alternate allele of a common missense variant at UGT1A4 (with a proline to threonine substitution, in high LD with the intronic lead variant and with reduced bilirubin binding) was associated with ~5-fold reduced risk for neonatal jaundice.
2. Further eQTL colocalization analyses showed association with UGT1A1 expression in the colon but not the liver—supporting a role for intestinal (rather than hepatic) UGT1A1 in neonatal jaundice.
3. Parental GWASs identified a third locus in ABO gene region in the maternal genome, consistent with a role for ABO incompatibility in neonatal jaundice. Further studies showed the relevance of transmitted (rather than non-transmitted) parental alleles on neonatal jaundice.
4. Genetic variants associated with adult jaundice were not associated with those identified in neonatal jaundice. However, PRS of adult bilirubin levels were strongly associated with neonatal jaundice risk, although this effect was driven by the UGT1A region in chromosome 2.

Strengths of this study is the novelty in conducting a GWAS of this clinically relevant issue of neonatal jaundice using a large, unique cohort of neonate/mother/father trio—with findings that were further replicated with potential physiological implications in intestinal vs hepatic expression, even though the findings related to UGT1A or ABO are not unexpected. Some limitations include the lack of diversity in using cohorts of Nordic European ancestry and absence of specific mechanistic or functional analyses, as the authors are aware. Could include a phenome-wide association study for the variants identified at least.

Response

We thank the reviewer for the constructive comments on our work.

We have now conducted replication analyses in neonates of African American (12,405 neonates, cases = 514) and European American (13,325 neonates, cases = 336) ancestries,

phenome-wide colocalization analyses with neonatal jaundice and adult bilirubin levels, evolutionary analyses to understand the differences in allele frequency of the *UGT1A4* missense variant and have modified the text for further clarity.

We agree that functional follow-up is desirable in any genome-wide association study and would be ideal also here for the UGT locus. We established new collaborations with experts in the field to test whether the *UGT1A4* missense variant identified in our GWAS could increase glucuronidation activity towards bilirubin of UGT1A4 enzyme in liver microsomes (to date, only UGT1A1 enzyme is known to play such a role). However, after facing several difficulties at the lab (among others, cells with the missense variant not growing, building new cell line from scratch, mass spectrometer going down for a couple of months, and bilirubin glucuronidation activity assays not working), we decided to halt the experiments and pursue them in a future project. In the present manuscript, we made sure to provide as much data as possible for other groups willing to perform functional follow-up, since we believe these experiments should be conducted in one way or another.

The analyses in individuals of predominantly African (rs6755571; OR = 0.50; p-value = 4.5×10^{-3} , allele frequency = 1.7%) and European (rs6755571; OR = 0.46; p-value = 1.2×10^{-5} ; allele frequency = 5.2%) ancestries replicate our main finding in the UGT genes region with concordant effect direction, despite the differences in allele frequency and outcome severity. We have added these analyses in the main results section, and in Supplementary Table 2. The small number of cases in our replication cohorts is a reflection of the uniqueness of the phenotype we are analyzing, meaning that data for replication analyses is not widely available. For instance, UK Biobank and FinnGen, with both > 400,000 subjects, include only 100 cases of neonatal jaundice each.

We have further added some clarifications with regard to the colocalization analysis of the missense variant with gene expression of *UGT1A1* in the colon. We observed such colocalization in two independent data sets – one from GTEx and another from CEDAR. Colon samples from these two data sets come from post-mortem (GTEx) and healthy individuals (CEDAR), suggesting the results are robust and replicable.

As suggested by the reviewer, we have also conducted phenome-wide colocalization analyses for the lead neonatal jaundice locus and the adult bilirubin levels locus in the *UGT* genes region with 952 phenotypes from the UK Biobank. This analysis provided further evidence that the neonatal jaundice and adult bilirubin loci at the *UGT* genes region are distinct, since only one of the colocalizing phenotypes (total cholesterol levels) overlapped the two sets (Response Figure 1). Interestingly, total cholesterol level has two conditionally-independent lead SNPs in the *UGT* genes region; one, the lead variant for neonatal jaundice, and the other, the lead variant for adult bilirubin levels). The total cholesterol level colocalizing in the UGT genes region is in line with the lower cholesterol levels observed in individuals with Gilbert's syndrome, which is characterized by a lifelong, mildly higher bilirubin levels caused by a lower glucuronidation activity of *UGT1A1* towards bilirubin. We have now added a new paragraph in the Results section, and these results can also be found in the Supplementary Figure 15 and Supplementary Table 5.

Response Figure 1. Posterior probability of colocalization between adult bilirubin levels and neonatal jaundice with 952 phenotypes from the UK Biobank at the UGT genes region. The x-axis shows the different phenotypes with a posterior probability of colocalization >0.9 with either adult bilirubin levels (left) or neonatal jaundice (right).

Reviewer #2 (Remarks to the Author):

We thank the reviewer for a careful assessment of our work. To ease the reviewing process, we have taken the liberty to split the reviewer's comments into numbered bullet points.

1. The authors have carried out a genome-wide association study (GWAS) of neonatal jaundice on a sample of 30,000 parents-offspring trios from Norway). Cases were aprox. 2000. Although the authors conclude that there are marked differences between the pathways involved in neonatal jaundice compared to those regulating bilirubin levels in adults, suggesting a distinct biological basis, these conclusions are not well based on the results. Similarly, the methodology used in the study to verify the proposed objective is not the correct one.

We agree with the reviewer that our phrasing was not very specific, and that we haven't identified different pathways. We have rephrased the conclusion and other statements about the differences in pathways to reflect that we have identified different genetic variants impacting the same pathway.

2. It is known that elevated bilirubin concentrations, primarily those genetically determined from childhood, are due to genetic variants in the UGT complex genes on chromosome 2. Specific polymorphisms can be very diverse depending on the geographical origin of the population studied. In this case it is a population from Northern Europe and the prevalence of genetic variants in these UGT1A* genes may be very different from the variants that can

be found in the population of Southern Europe. There may still be more differences in genetic variants in UGT genes if non-European populations are analyzed.

In the introduction, we do acknowledge that rare genetic mutations in *UGT1A1* are known to affect the risk of neonatal jaundice. However, the effect of common variants has not been investigated before using a hypothesis-free approach. In fact, while many candidate gene studies have focused on variants within the UGT genes complex, we haven't been able to identify a single study assessing the effects of rs6755571 on neonatal jaundice. Also, none of the previous candidate gene studies pass the genome-wide significance threshold (5×10^{-8}) that is currently a standard in genetic studies, and that has contributed to a wealth of true positive associations in the field. In this regard, our study shows no associations between genetic variants in *SLCO1B1* gene and neonatal jaundice, while several candidate gene studies, with low sample sizes, have found such associations, although using nominal p-value thresholds ($p < 0.05$). Well powered studies, as the one presented here, are required to avoid spurious associations.

We haven't found large differences in the allele frequency of the rs6755571 allele in other European ancestries, including samples from Southern Europe (see Response Figure 2). However, we agree with the reviewer that the allele frequency is different in populations of non-European ancestries. We have now replicated our top locus in the *UGT1* genes region in individuals of African and other European ancestries (see response to the reviewer 1). The A allele of the missense variant is almost non-existent in individuals of East Asian ancestry, so a GWAS in this population to replicate our finding is not doable. We nonetheless conducted several evolutionary analyses to understand potential differences in the frequency of the haplotype containing the rs6755571 A allele in different ancestries, suggesting that the region has undergone recent positive selection that has shortened the haplotype around this variant, but the results were not statistically significant.

We have added the results from the evolutionary analyses in the main text and can be found in Supplementary Figure 5 and Supplementary Table 4.

3. The protective variant highlighted by the authors as protective (rs6755571, A allele OR = 0.2, value = 2.7×10^{-55} , frequency = 6.4%, may be a variant with a great founder effect in such a population, so additional GWAS in diverse populations (Southern Europe, Hispanic, Asian, African, etc.) are needed to bring more knowledge about the genetics of bilirubin concentrations in children and adults and support the author's conclusions.

We do not consider our discovery or replication samples to come from a bottlenecked population, and so founder effects are not generally expected. This is in line with the allele frequency of our lead SNP being relatively common in other European ancestries (see Response **Figure 2**). As mentioned above, we have now included results from a GWAS in African and European ancestries, replicating our results, and with the same effect direction.

There are no major differences in allele frequency in individuals of other European ancestries, as seen in exome sequencing data from the Regeneron Genetic Center (accept terms and search for "2:233718890:C:A"), which holds data from 821,979 unrelated individuals (Response **Figure 2**). In this same database, the frequency of this allele is >1%

in all ancestries, except in individuals of South East Asian ancestry, which was already mentioned in our manuscript.

Ancestry	AC	AN	AAF*
All (ALL)	84413.00	1643948.00	0.051
> African (AFR)	1689.50	98816.27	0.017
> South Asia (SAS)	1289.74	81336.78	0.016
> East Asian (EAS)	337.41	30011.15	0.011
East Asia (E.ASIA)	300.42	22571.27	0.013
South East Asia (SE.ASIA)	36.99	7439.88	0.005
> European (EUR)	75807.15	1261224.80	0.060
> British Isles (BI)	66968.09	1044705.70	0.064
> Northern Europe (N.EUR)	1339.13	34496.63	0.039
> Central Europe (C.EUR)	3859.94	84621.90	0.046
> Southern Europe (S.EUR)	3639.98	97400.57	0.037
> Indigenous American (IAM)	-	-	-
Middle East (MEA)	340.87	7806.56	0.044

Response Figure 2. Frequency of the A allele of the 2:233718890:C:A missense variant in the Million Exome Variant Browser from the Regeneron Generic Center in individuals of different ancestries.

The allele frequency from the Regeneron Genetic Center data is largely consistent with data from gnomAD - allele frequency is >1% in individuals of any ancestry except in individuals of East Asian ancestry, in whom the allele is almost nonexistent (Response **Figure 3**). In line with this, only 1 such allele was observed among 54,301 individuals from the Tohoku Medical Megabank Project in Japan.

In light of this, we made the following changes to the manuscript:

- we have added the replication analyses and allele frequency in neonates of African and European American ancestries (main text and Supplementary Table 2)
- we have added the allele counts from the Tohoku Medical Megabank Project
- we have rephrased the text for clarity

Genetic Ancestry Group Frequencies

Note: Local ancestry data is available for this variant by selecting the tab below. See our blog post on local ancestry inference for Latino/Admixed American samples in gnomAD for more information.

gnomAD	HGDP	1KG	Local Ancestry																																																																						
   Genetic Ancestry Group Allele Count Allele Number Number of Homozygotes Allele Frequency     > European (non-Finnish) 74784 1179912 2470 0.06338   > Middle Eastern 300 5996 12 0.05003   > Remaining 3079 62480 99 0.04928   > Ashkenazi Jewish 1164 29604 25 0.03932   > Admixed American 1825 60028 39 0.03040   > European (Finnish) 1724 64040 19 0.02692   > African/African American 1225 75046 10 0.01632   > Amish 14 912 0 0.01535   > South Asian 1028 91068 19 0.01129   > East Asian 5 44882 0 0.0001114   XX 44109 812336 1417 0.05430   XY 41039 801632 1276 0.05119   Total 85148 1613968 2693 0.05276   				Genetic Ancestry Group	Allele Count	Allele Number	Number of Homozygotes	Allele Frequency	> European (non-Finnish)	74784	1179912	2470	0.06338	> Middle Eastern	300	5996	12	0.05003	> Remaining	3079	62480	99	0.04928	> Ashkenazi Jewish	1164	29604	25	0.03932	> Admixed American	1825	60028	39	0.03040	> European (Finnish)	1724	64040	19	0.02692	> African/African American	1225	75046	10	0.01632	> Amish	14	912	0	0.01535	> South Asian	1028	91068	19	0.01129	> East Asian	5	44882	0	0.0001114	XX	44109	812336	1417	0.05430	XY	41039	801632	1276	0.05119	Total	85148	1613968	2693	0.05276
Genetic Ancestry Group	Allele Count	Allele Number	Number of Homozygotes	Allele Frequency																																																																					
> European (non-Finnish)	74784	1179912	2470	0.06338																																																																					
> Middle Eastern	300	5996	12	0.05003																																																																					
> Remaining	3079	62480	99	0.04928																																																																					
> Ashkenazi Jewish	1164	29604	25	0.03932																																																																					
> Admixed American	1825	60028	39	0.03040																																																																					
> European (Finnish)	1724	64040	19	0.02692																																																																					
> African/African American	1225	75046	10	0.01632																																																																					
> Amish	14	912	0	0.01535																																																																					
> South Asian	1028	91068	19	0.01129																																																																					
> East Asian	5	44882	0	0.0001114																																																																					
XX	44109	812336	1417	0.05430																																																																					
XY	41039	801632	1276	0.05119																																																																					
Total	85148	1613968	2693	0.05276																																																																					

Response Figure 3. *Frequency of A allele of the rs6755571 missense variant in the gnomAD variant browser in individuals of diverse ancestries.*

4. It is also known that these additional genes not located on chromosome 2 may be different depending on the population analysed. The authors would therefore need to analyze more diverse adult populations to better quantify the most relevant genes in each geographical origin and not just focus on populations in Northern Europe. With this analysis they could verify that the genetic risk scores (basically the non-UGT genes) are heterogeneous in the adult population and can in turn be modulated by lifestyle variables.

The analysis of adult bilirubin levels in Europeans and other diverse ancestries has been the focus of other publications (Sinnott-Armstrong, *Nat Genet*, 2021; Kanai, *Nat Genet*, 2018; Chen, *Cell Genomics*, 2023). We want to point out that biology is not different across ancestries, and that the only difference that can arise in the observation or not of genetic associations is largely due to differences in allele frequency and linkage disequilibrium between ancestries. We consider the study of gene-environment interactions of adult bilirubin levels to be out of the scope of our study.

5. Similarly, in the study of children, the authors may have analyzed genetic modulations through other factors of parental style and in pregnancy to provide new information on new interactions with genetic variants in that population.

In the manuscript we show no interactions of the lead SNP with maternal-fetal ABO blood group incompatibility, and a small interaction with gestational duration (interaction p-value = 6.2×10^{-4}). Our analysis was limited to two of the main contributors to neonatal jaundice. We want to point out that the study of subgroups (gene-environment interactions) typically requires several times larger overall sample size to reach sufficient power, and therefore is rarely included as part of discovery GWASs.

6. Another limitation is that the authors perform functional analyses in silico. This implies limitations on data sources and conclusions. It would have been more appropriate to do some expression analysis in the wet lab. In general, the article has significant limitations and more analyses are needed in other diverse populations and with better methodology.

We refer the reviewer to the response to reviewer 1 for the lack of functional follow-up. We want to highlight that we have used previously published data to perform our in silico analysis, but these data included gene expression (RNA sequencing) analysis in the lab. While these experiments were not conducted in the colon or liver of neonates, we consider our results robust and replicable: the colocalization we observe appears in two independent data sets – one in colon samples from post-mortem (GTEx) and another from healthy (CEDAR) individuals. We have now clarified this in the result section.

REVIEWERS' COMMENTS

Reviewer #1 (Remarks to the Author):

The revision addressed most of the concerns raised (except the functional/mechanistic analyses which were attempted but not successful) and provide new insights regarding genetic risk for neonatal jaundice from a unique Nordic cohort-with further replication in other ancestries. One comment is to modify the sentence in line 373-374 in Discussion, as additional populations have been examined in this revision.

Reviewer #3 (Remarks to the Author):

The manuscript titled "Genome-wide analyses of neonatal jaundice reveal a marked departure from adult bilirubin metabolism" by Pol Solé-Navais et al. has already undergone one round of reviews, receiving valuable feedback from reviewers. These previous reviews have highlighted the strengths and weaknesses of the study, providing constructive criticism that has undoubtedly contributed to the refinement of the work. This review, therefore, builds upon the foundation laid by earlier assessments.

This study represents the first GWAS focused on neonatal jaundice, encompassing nearly 30,000 parent-offspring trios from Norway, with approximately 2,000 cases of jaundice, revealing key genetic insights.

The primary aspects of the manuscript, that were modified based on the reviewers' suggestions include:

1. The identification of a significant genetic locus within the UGT1A gene region. A missense variant in UGT1A4 was found to reduce the risk for neonatal jaundice by fivefold.

The strength of this association was reaffirmed through replication in separate neonatal cohorts of African American and European ancestry, which addressed reviewers' concerns about genetic diversity not being sufficiently represented, given the initial focus on the Northern European population. The authors also included allele frequency in neonates of African and European American ancestries, as well as allele counts from the Tohoku Medical Megabank Project and evolutionary analyses for rs6755571.

2. Colocalization analysis between neonatal jaundice and cis-eQTLs across various tissues and cell types (eQTL Catalogue) was performed, revealing a correlation between the identified genetic variant and UGT1A1 expression specifically in the colon, rather than the liver, which suggests a tissue-specific regulatory effect. Following reviewers' suggestions, they replicated the findings using two independent eQTL datasets: one from healthy adults (CEDAR) and another from post-mortem tissues (GTEx), which strengthened the robustness of their results.

3. As requested, the authors included phenome-wide colocalization analyses (using 952 phenotypes from the UK Biobank) for the lead neonatal jaundice locus and the adult bilirubin levels locus within the UGT genes region, reaffirming the distinct nature of the neonatal jaundice and adult bilirubin loci.

4. One of the main concerns raised by the reviewers, which I share, is the lack of functional evidence to support the obtained results. The authors have not addressed this issue, but they provided a rationale for this gap. Although addressing this gap is critical, the manuscript nonetheless presents new perspectives in the field of neonatal jaundice and contributes to providing new clues to potential underlying mechanisms.

Overall, I find that the authors have significantly improved the manuscript by implementing all the recommendations provided by the reviewers concerning the genetic and in silico analyses. Their revisions have resulted in a clearer and more cohesive text. Despite the lack of functional evidence, I still maintain that the manuscript remains valuable and intriguing, offering fresh insights into the field of neonatal jaundice.

REVIEWERS' COMMENTS

Reviewer #1 (Remarks to the Author):

The revision addressed most of the concerns raised (except the functional/mechanistic analyses which were attempted but not successful) and provide new insights regarding genetic risk for neonatal jaundice from a unique Nordic cohort-with further replication in other ancestries. One comment is to modify the sentence in line 373-374 in Discussion, as additional populations have been examined in this revision.

Response

We thank the reviewer for the careful revision of our work. We have now modified the sentence in lines 373-374, which now reads as:

It is also unclear why the alleles observed here with large protective neonatal effects – thus presumably highly selected for – are relatively uncommon, and almost absent in some populations. More broadly, the prevalence of missense variants with regulatory effects on genes distinct from the one it encodes is not well established and may induce artifacts in variant-to-gene mapping efforts.

Reviewer #3 (Remarks to the Author):

The manuscript titled "Genome-wide analyses of neonatal jaundice reveal a marked departure from adult bilirubin metabolism" by Pol Solé-Navais et al. has already undergone one round of reviews, receiving valuable feedback from reviewers. These previous reviews have highlighted the strengths and weaknesses of the study, providing constructive criticism that has undoubtedly contributed to the refinement of the work. This review, therefore, builds upon the foundation laid by earlier assessments.

This study represents the first GWAS focused on neonatal jaundice, encompassing nearly 30,000 parent-offspring trios from Norway, with approximately 2,000 cases of jaundice, revealing key genetic insights.

The primary aspects of the manuscript, that were modified based on the reviewers' suggestions include:

1. The identification of a significant genetic locus within the UGT1A gene region. A missense variant in UGT1A4 was found to reduce the risk for neonatal jaundice by fivefold.

The strength of this association was reaffirmed through replication in separate neonatal cohorts of African American and European ancestry, which addressed reviewers' concerns about genetic diversity not being sufficiently represented, given the initial focus on the Northern European population. The authors also included allele frequency in neonates of African and European American ancestries, as well as allele counts from the Tohoku Medical Megabank Project and evolutionary analyses for rs6755571.

2. Colocalization analysis between neonatal jaundice and cis-eQTLs across various tissues and cell types (eQTL Catalogue) was performed, revealing a correlation between the identified genetic variant and UGT1A1 expression specifically in the colon, rather than the liver, which suggests a tissue-specific regulatory effect. Following reviewers' suggestions, they replicated the findings using two independent eQTL datasets: one from healthy adults

(CEDAR) and another from post-mortem tissues (GTEx), which strengthened the robustness of their results.

3. As requested, the authors included genome-wide colocalization analyses (using 952 phenotypes from the UK Biobank) for the lead neonatal jaundice locus and the adult bilirubin levels locus within the UGT genes region, reaffirming the distinct nature of the neonatal jaundice and adult bilirubin loci.

4. One of the main concerns raised by the reviewers, which I share, is the lack of functional evidence to support the obtained results. The authors have not addressed this issue, but they provided a rationale for this gap. Although addressing this gap is critical, the manuscript nonetheless presents new perspectives in the field of neonatal jaundice and contributes to providing new clues to potential underlying mechanisms.

Overall, I find that the authors have significantly improved the manuscript by implementing all the recommendations provided by the reviewers concerning the genetic and in silico analyses. Their revisions have resulted in a clearer and more cohesive text. Despite the lack of functional evidence, I still maintain that the manuscript remains valuable and intriguing, offering fresh insights into the field of neonatal jaundice.

Response

We thank the reviewer for the positive assessment of our work and appreciate the understanding of the value of our work despite the lack of functional follow-up.